# Long-Term Outcome and Quality of Life in Patients Treated for Head and Neck Sarcoma during Pediatric Age: Considerations from a Series of 4 Cases

**DOI:** 10.3390/reports6010016

**Published:** 2023-03-17

**Authors:** Martina Buchignani, Arrigo Pellacani, Sara Negrello, Mattia Di Bartolomeo, Monica Cellini, Alessia Pancaldi, Alessandra Buttafoco, Camilla Migliozzi, Lorenzo Iughetti, Luigi Chiarini, Alexandre Anesi

**Affiliations:** 1Cranio-Maxillo-Facial Surgery Unit, Department of Medical and Surgical Sciences for Children & Adults, University of Modena and Reggio Emilia, Largo del Pozzo 71, 41124 Modena, Italy; 2Unit of Dentistry and Maxillo-Facial Surgery, Surgery, Dentistry, Maternity and Infant Department, University of Verona, P.le L.A. Scuro 10, 37134 Verona, Italy; 3Cranio-Maxillo-Facial Surgery Unit, University Hospital of Modena, Largo del Pozzo 71, 41124 Modena, Italy; 4Pediatric Onco-Haematology Unit, Department of Medical and Surgical Sciences for Children & Adults, University of Modena and Reggio Emilia, Largo del Pozzo, 71, 41124 Modena, Italy; 5Unit of Psychology, Azienda Ospedaliero Universitaria di Modena, Largo del Pozzo 71, 41124 Modena, Italy; 6Pediatric Unit, Department of Medical and Surgical Sciences for Children & Adults, University of Modena and Reggio Emilia, Largo del Pozzo 71, 41124 Modena, Italy

**Keywords:** sarcoma, pediatric, HRQoL, head and neck cancer, survival

## Abstract

Pediatric sarcomas are rare malignancies accounting for about 10% of solid tumors in childhood. Sarcomas represent a heterogeneous group of malignancies, especially they include Rhabdomyosarcoma (RMS) and Non-Rhabdomyosarcoma Type Sarcomas (STSNR). Therapeutic management of pediatric sarcomas is challenging. Multidisciplinary approach including chemotherapy, surgery and radiotherapy is the treatment of choice. The correct management of affected patients can be demanding, especially in terms of preserving a good quality of life (QoL). In the present study we described our experience with a series of patients diagnosed with head and neck non-metastatic sarcoma between January 1995 and December 2020, referred to Pediatric Oncology or to Maxillo-Facial Surgery of University Hospital of Modena and Reggio Emilia.The survivors were asked to complete questionnaires on health-related quality of life (HRQoL; PedsQL and EORTC QLQ-H&N 43). We found satisfactory results in terms of global HRQoL; although outstanding issues persist, such as deterioration in masticatory function, and speech. Long-term sequelae of multimodality treatment may lead to both significant aesthetic-functional and psychosocial implications that need to be further investigate during the whole the care pathway.

## 1. Introduction

Malignant tumors of the head and neck are rare entities in pediatric patients, representing 12% of all childhood malignancies. Altogether, cancers constitute the second leading cause of death in children and adolescents, following trauma [1].

Among these pediatric tumors, sarcomas are quite rare, so the correct management of affected patients can be challenging, especially in terms of preserving a good quality of life (QoL).

Sarcomas represent a heterogeneous group of malignancies accounting for about 10% of solid tumors in childhood [2]. Sarcomas (PSs) are classified into two main categories: (1) soft tissue sarcomas (STSs) and (2) bone sarcomas (BSs) [3].

Soft tissue sarcomas, accounting for 8% of all pediatric cancers, include Rhabdomyosarcoma (RMS) and Non-Rhabdomyosarcoma Type Sarcomas (STSNR) [4]. Epidemiologically, the former is certainly the most common soft tissue sarcoma in the pediatric population [5], with 4.5 cases per million children worldwide (about 350 cases per year), half of the cases recorded in the first decade of life [6].

Head and neck onset rhabdomyosarcomas are the most common (36%), followed by the urogenital system (23%) and extremities (19%) forms [7].

It should be noted that in 20% of RMS patients metastases are already present at the time of diagnosis, clinically or radiologically detectable. Lymph nodes, lungs and bones are habitually involved [8]. Together with the patient age, this is probably the most impacting prognostic factor on long-term survival [9].

Similar to rhabdomyosarcoma, STSNRs arise from primary mesenchymal tissue [10]. This subset of sarcomas has recently been increasing over time, reaching an estimated incidence of about 250–300 cases per year in the United States, thus representing approximately 50% of newly diagnosed STSs, mostly in adolescents and young adults [11].

Bone sarcomas are rare neoplasms with an overall incidence of 0.3 cases per 100,000 per year, and among them, osteosarcoma is the most common histological type with a peak of 0.8 cases per 100,000/year in the 15–19 age group and a slight prevalence in males. Mainly affecting the long bones, osteosarcoma is rarely found in the head and neck (10%), where maxilla and mandible are the mostly affected sites [12]. However, maxillofacial osteosarcomas would seem to present a better prognosis than other anatomical sites, with survival rates of about 80%. This is probably due to the lower tendency to metastasize, with a frequency of 17% compared to 80% of long bone sarcomas [13].

Ewing’s sarcoma (EWS) is a separate nosological entity from STSNRs group, although it is formally classified within bone sarcomas. Indeed, Ewing’s sarcoma is defined as a malignant primary neuroectodermal tumor (PNET) that recognizes multiple variants based on the site of onset, all resulting from undifferentiated mesenchymal or neuroectodermal stem cells of bony or soft tissues [14]. With an incidence of 1.5 cases per million worldwide, EWS is the second most frequent primary bone neoplasm, with the majority of cases recorded before the age of twenty [15].

Representing 80% of all reported cases, primary bone EWS mainly affects the lower limbs (41%), followed by the pelvis (26%), chest wall (16%), upper extremity (9%), spine (6%) and cranium (2%). Maxillofacial forms are extremely rare (4–9% of all cases), predilecting the mandible and the skull base, then the orbit and nasal cavities [15].

Clinically, head and neck sarcomas usually present as rapidly growing masses, with or without pain, paraesthesia, trismus, facial paralysis, or nasal discharge [15]. In most cases, the symptomatology is due to contiguous bony or neurovascular structures compression, pushing or displacing resulting from tumor mass effect [16]. Trigeminal impairment, headache, and increased intracranial pressure are common symptoms that may occur as well [15].

The diagnosis of pediatric sarcomas is based on the integration of clinical, radiological and histopathological findings. The biopsy of the lesion is needed for histological confirmation but should represent the conclusion of the investigation process.

Regarding head and neck sarcomas, both CT and MRI are central in disease staging. PET scan may assist in identifying locoregional or distant secondary localizations [6].

Therapeutic management of pediatric sarcomas is challenging. Aiming at the complete eradication of the disease, a multidisciplinary approach include chemotherapy, surgery and radiotherapy [6]. Timing and intensity of these therapeutic options should be based on individual prognostic variables, such as patient age, tumor site and size, nodal status and metastases [6]. On the other hand, acute and long-term side effects are not negligible [16]. Pain, nausea and vomiting, as well as acute bone marrow toxicity are common during antineoplastic treatment. Among the most worrying long-lasting deficits, chronic renal failure and infertility have been reported [17].

Several studies have shown how health-related quality of life (HRQoL) might be negatively influenced by therapies implications [16]. This is especially true for head and neck pediatric sarcomas. Given the sensitive and impetuous psychomotor development at this stage of life, any functional or aesthetic impairment may heavy impact on child global health. Long-term survived patients frequently complain of their unsatisfactory appearance, severely limiting daily social, work, and relational activities. Proper and targeted psychological support, both for the patient and the family, is therefore crucial.

The improvement of therapeutic strategies in recent years, reflected in increasingly positive disease-free survival data, confirms the need to protect post-treatment quality of life.

However, HROoL data of sarcomas pediatric patients reported in the literature are sparse. Clearly, this is also due to the rarity of these malignancies. Furthermore, it should be emphasized that data related to children’s symptoms may not reflect the real patient experiences, as they are routinely documented by clinicians. For these reasons, standardized HROoL questionnaires have recently been developed to support healthcare professionals in tailoring an individual based holistic management [18,19].

Our experience in the multidisciplinary management of patients with rare head and neck sarcomas over a long period of 25 years is reported here. The evaluation of long-term survival, together with the post-treatment quality of life are the main goals on which the authors focused.

## 2. Materials and methods

All patients included in this study were diagnosed with sarcoma of head and neck during the childhood period.

Patients were selected based on a diagnosis of head and neck non-metastatic sarcoma in the period January 1995 and December 2020, referred to Pediatric Oncology or to Maxillo-Facial Surgery of University Hospital of Modena and Reggio Emilia. We excluded patients lacking an adequate period of follow-up (minimum one year follow-up).

For the study purposes, medical records of patients were retrospectively reviewed and data regarding gender, age, treatment protocol, surgical approach and outcome were collected.

Patients eligible for the present study were asked to complete the Pediatric Quality of Life Inventory (PedsQL) and EORTC QLQ-H&N 43. These HRQoL questionnaires were administered to patients during the period from October 2020 to April 2021.

PedsQL 4.0 Generic Core Scales is a validated modular tool consisting of 23 items assessing HRQoL on four sub-scales: physical functioning, emotional functioning, social functioning, and school/work functioning. The questionnaire refers to children and adolescent between 2- and 18-year-old. Scores ranged 0 to 100, higher scores indicate better quality of life in the specific domain.

EORTC QLQ-H&N 43 is a validated questionnaire consisting of 43 questions divided into 6 multi-item scales and 13 single-item scales. It is validated for adult patient only. Consequently, a specific clinical survey form for the pediatric and/or adolescence population administered by a pediatric hematologist/oncologist, on the basis of the same domains investigated by EORTC QLQ-H&N 43. Obviously, this specific form excludes the sexuality domain. The scores are then linearly transformed into a scale from 0 to 100, where 0 represents the absence of problems, while 100 represents the maximum impairment.

Addictionally, in this study we evaluated the 5-year and 10-year Overall Survival (OS) after sarcoma diagnosis. OS was defined as the interval between diagnosis and either death for any cause or the last follow-up.

The study was approved by the local institutional review board of the University Hospital of Modena (project identification code 1/2021). Informed consent for the study participation was obtained from all patients or their legal guardians in accordance with the Declaration of Helsinki.

## 3. Case Presentation Section

### 3.1. Case 1

A 6-year-old male patient presented to Maxillo-Facial Surgery in June 2017 with painful maxillary swelling.

Biopsy was performed by the dentist and the diagnosis of Ewing Sarcoma was established. FISH analysis showed the presence of EWSR1 gene rearrangement in 11.7% of the cells examined in the specimens.

MRI scan was performed, showing an extensive infiltrative mass of 4 × 2.9 × 3.6 cm extending into right maxillary sinus. No cervical lymphadenopathy was reported (Figure 1). 

The patient underwent neoadjuvant high-dose chemotherapy with Etoposide, Ifosfamide, Vincristine, Adriamycin, Cyclophosphamide (according to AIEOP ISG/SSG III standard risk-localized protocol) followed by stem-cell rescue. Subsequently, a right maxillectomy (Figure 2) according to Weber-Fergusson approach and immediate reconstruction of the soft palate with Bichat fat pad flap was performed.

Due to positive surgical margins, the patient was eligible for chemotherapy in adjuvant-setting. Moreover, second-step surgery was performed to reach oncological radicality. As required by the AIEOP ISG/SSG III protocol for non-metastatic Ewing’s family tumors, adjuvant radiation therapy was delivered (50.4 Gy).

4-year clinical follow-up showed no signs suggestive for disease recurrence (Figure 3).

### 3.2. Case 2

A 3 years-old male referred to Pediatric Oncology in January 2014 complaining of superior eyelid edema and bilateral exophtalmus. The patient was in good general condition. The MRI study showed a naso-ethmoidal mass with intraorbital extension. There were also bony defects in the right medial orbital wall and floor. Additionally, the contralateral orbit was also partly involved (Figure 4).

A surgical biopsy was performed, and histological investigation confirmed the diagnosis of Ewing Sarcoma. EWSR1 gene rearrangement was identified in 70% of cells.

The patient started a ten-weeks induction chemotherapy (Ifosfamide, Etoposide, Vincristine, Actinomycin D, Adriamycin, Cyclophosphamide) according to the ISG/SSG III protocol for non-metastatic Ewing’s family tumor.

Post-chemotherapy endoscopic middle turbinectomy and anterior ethmoidectomy was carried out. Post-operative CT scan showed an involvement of medial orbital wall. Accordingly, the patient received high-dose chemotherapy followed by stem-cell rescue, and a post-operative radiotherapy (total dose 50.4 Gy).

At 6-years follow up no recurrence of disease was observed (Figure 5).

### 3.3. Case 3

In July 2008, a 5-years-old Albanian female presented to the Pediatric Oncology with a progressive swelling of the left cheek. The patient underwent surgical biopsy in Albania resulting in alveolar rhabdomyosarcoma diagnosis.

MRI showed a wide RMS (7.5 × 6 × 5 cm) involving the left masticatory space with maxillary sinus invasion. Left cheek and orbital floor were affected as well, and bilateral positive lymph nodes were detected (Figure 6).

Treatment consisted of neo-adjuvant chemotherapy with Ifosfamide, Vincristine, Actinomycin D and Doxycycline, according to EpSSG RMS protocol for non-metastatic rhabdomyosarcoma. After chemotherapy, second-look radical surgery was performed. The anatomopathological examination of the surgical specimen confirmed the surgical radicality. Then, adjuvant radiation therapy was administered (total dose of 52.64 + 50.4 Gy). She completed the treatments with 4 cycles of Ifosfamide, Vincristine, Actinomycin D. Last clinical-radiological follow-up showed no disease relapse. Multiple autologous lipotransfer procedures have been performed during the last two years, and satisfactory aesthetic outcome was obtained (Figure 7).

### 3.4. Case 4

In January 2004, a 6-year-old patient was referred to the Pediatric Oncology with a right mandibular lymphadenopathy developed over a month. Clinically, he had no response to antibiotics. MRI scan highlighted a 6.5 × 5.7 × 4.5 cm mass in the right masticatory space involving right pterygoid muscles, right temporal fossa, right masseter, and the mandibula ramus. The biopsy reported for alveolar rhabdomyosarcoma. CT scan showed a 5 mm pulmonary nodule in the right superior lung lobe, thus the neoplasm was initially diagnosed as metastatic.

A first chemotherapic protocol reserved to metastatic patients was therefore administered. After consultation with the experts of the Disease Protocol Referent Centre, and based on the excellent response of the patient to the treatment, the pulmonary imaging finding was considered as non-pathological. This led us to change the therapy to EpSSG RMS 2004 chemo-radiotherapy protocol. Post-treatment imaging confirmed the complete disease remission.

Unfortunately, in February 2005 RMI scan showed the onset of a 13 × 16 mm mass in the right masticatory space and a disease recurrence was diagnosed (Figure 8). The patient underwent radical surgery, and subsequently he received a 16-week adjuvant chemo-radiation therapy (4 cycles of Topotecan and Cyclophosphamide and total dose of 45 Gy). No positive surgical margins were found on definitive histopathological examination.

Due to the unfavorable features of the tumor, the patient underwent the maintenance treatment with Cyclophosphamide and Vinorelbine for a 24-week period. He is now free from disease at 15 years follow up (Figure 9).

## 4. Results

During the observational period, only 6 patients were eligible for our study. One patient died due to disease progression. A second patient was lost to follow-up. The median age at the diagnosis was 4.93 years.

The overall survival, both at 5-year and at 10-year was 83.34% (Appendix A).

Eligible patients included three male and only one female. Patient demographics are summarized in Table 1.

In the present study, three patients were diagnosed with rhabdomyosarcoma, two had Ewing’s sarcoma, while only one patient had non-rhabdomyosarcoma type soft tissue sarcoma, specifically fibrosarcoma.

Most of the tumor localizations (50%) involved the pterygopalatine fossa and/or the infratemporal fossa with extension to the parapharyngeal space; the other areas of prevalent tumor involvement were the nasal cavities and paranasal sinuses (25%) and the hard palate with relative maxillary alveolar arch (25%). At the time of diagnosis, 50% of patients had a tumor extension greater than 5 cm (Table 2).

### 4.1. Quality of Life Results

Patients eligible for this study were asked to complete the Pediatric Quality of Life Inventory (PedsQL) and the EORTC QLQ-H&N 43.

### 4.2. PedsQL 4.0

The PedsQL 4.0 Generic Core Scale “self-report” was administered to three patients according to the different age version of the questionnaire. One patient was not administered with PedsQL 4.0 because he was 18-year-old at time of investigation.

The Social functioning domain reached the higher score (95 ± 8.66). Physical health and school functioning had an average score of 84.37 ± 8.27 and 83.33 ± 12.58, respectively. Psychosocial health domain achieved an average score of 82.44. The results are reported in Table 3.

### 4.3. EORTC QLQ-H&N43

The EORTC QLQ-H&N43 questionnaire was administered to all the population included in the study. Three patients were submitted to the clinical investigation form adapted to the paediatric-adolescent population. Only one patient was administered with the questionnaire in the original form.

Problems with teeth (33.33 ± 39.5) and speech (25 ± 25.75) were the most underrate multi-item domain in the study population. Therefore, body image and fear for progression received a score of 16.66 ± 26.45 and 12.5 ± 8.33 respectively (Table 4). Sexuality domain had been considered only in patients that had reached adulthood at the time of investigation.

On the single-item rating scales, below average scores were found in problems opening the mouth (50.00 ± 33.33) and concern for social interactions (16.67 ± 33.33), as seen in Table 5. No significant problems have been highlighted in the other domains.

## 5. Discussion

Pediatric head and neck sarcomas are rare malignancies. Therefore, they are particularly challenging to treat due to their low incidence and nonspecific clinical appearance.

Surgery plays a critical role in the treatment of head and neck sarcomas. In the last four decades, multimodality treatment such as multiagent chemotherapy, possibly combined with radiotherapy, resulted in a significant improvement of long-term survival [20]. In the present study, we observed an overall survival both at 5-years and 10-years of 83.34%. Survival findings are consistent with the literature [21,22,23], even if they have no statistical validity.

Complete surgical resection is beneficial for HNRMS and is an important aspect of the multimodal management protocol endorsed by the COG. However, depending on tumor site, complete tumor resection can be challenging. It is crucial that the patients are referred to a center with high experience in pediatric oncology surgery [24,25].

Although intensive treatment regimens significantly improve survival, they could often lead to long-term undesirable consequences that drastically affect both physical and psychosocial features of patients. So et al. [26] suggest that despite the relevant improvement in morbidities in pediatric population, there are certain long-term side effect of treatments such as xerostomia, social eating, fatigue, and physical functioning, produced in part to surgery but largely from radiotherapy. Indeed, radiation therapy in the pediatric patient involves bone and muscle structures that are not yet completely formed. On the other hand, radiotherapy techniques have improved greatly in recent years, particularly in terms of reducing side effects. Thus, irradiation is currently the gold standard for local disease control. All these aspects should be regularly monitored over time. So far, there is a lack of studies in the literature that investigate the long-term consequences and the quality of life in patients treated during the pediatric age. Regarding this aspect, our study is one of the first that examines the HRQoL in patient treated for sarcoma of head and neck.

We believe that this analysis can offers an insight for assessing psychosocial well-being in head and neck sarcoma survivors.

Together with early diagnosis of disease recurrence, the purpose of patient monitoring over time is also to detect possible unfavourable effects of therapies. This allows to plan specific interventions, such as second-step reconstructive surgery or multidisciplinary rehabilitation strategies, to drastically limit both functional and psychological sequalae.

In our patients we observed that the quality of life is preserved. The mean overall score of the PedsQL 4.0 Generic score “self-report” was 82.92 ± 10.67, while the overall psychosocial health score was 82.44 ± 13.02.

Our result differs from those reported by Eiser et al. [27], who did not find psychosocial health preservation in long-surviving pediatric patients.

Although our data seems to be encouraging, it is necessary to consider the small number of patients enrolled in the present study, as well as the socio-demographic and age variability not allowing for generalizations.

It is essential to pay attention to the domains of the PedsQL 4.0 that received the lowest overall score. Among these, below-average score was observed in the emotional health domain of PedsQL 4.0, although there was variability in responses (SD = 22.5). We hypothesized that this finding may have been impacted by the COVID-19 pandemic. Certainly, the lockdown may have played an important role in increasing anxiety and uneasiness, by radically changing daily routines [28,29,30,31,32,33,34].

In contrast with Vaarwerk et al. [35], our data reports that school functioning domain seems to be preserved. The authors recorded HNRMS survivors impaired scores regarding school/work functioning, and a negative impact in social interaction due to their aesthetic appearance. Results from the PedsQL 4.0 and EORTC QLQ-H&N 43 about social functioning and social-eating seems to be paired in our patient series.

According to the score of EORTC QLQ H&N43, masticatory and phonatory function are often impaired even after several years after treatments. It is mandatory to consider that surgical approaches, and radiotherapy had been performed during the developmental age. Crucial functions such as swallowing, senses of taste and smell, speech, and mastication (teeth, saliva, proper occlusion), can be affected or lost irretrievably. As expected, our results highlight that dental problems (33.33 ± 39.5), difficulty in oral opening (50.00 ± 33.33) and subsequent speech difficulty (25 ± 25.75) are all domains showing severe functional impairment. 

Furthermore, we submitted questionnaire concerning QoL years after the end of treatments. This provides a delayed sight in the patient’s perception of the long-term consequences of sarcoma and its influence on daily routine; thus, we do not have any information to compare the immediate after-treatment period.

It is interesting to notice that fear of disease progression (EORTC QLQ-H&N 43) seems to be not particularly affected in terms of scoring. Based on our experience, this could be due to the patient perception of caring resulting from a long-term, multidisciplinary and systematic monitoring program. In our opinion, this result could be referred to tailored interventions such as psychosocial care or reconstructive surgery. Moreover, the family background and the patient social network and support, although not investigated in the present study, might also justify this result as a psychological and emotive protective factor. Therefore, this domain should be more specifically investigated by further studies.

## 6. Conclusions

Head and neck sarcomas are rare malignancies in pediatric population.

According to the international literature, we found positive long-term disease-free survival rate, with an overall survival at 5-years and 10-years from diagnosis equal to 83.34%, respectively.

Long-term sequelae of multimodality treatment may lead to both significant aesthetic-functional and psychosocial implications. Specific questionnaires we administered to patients included in the present study showed satisfactory results in terms of HRQoL. We hypothesize this favourable outcome depends on the proper multidisciplinary patient management, and on the tailored treatment strategies as well. These results require to further investigate the patient perception of the entire care pathway.

## Figures and Tables

**Figure 1 reports-06-00016-f001:**
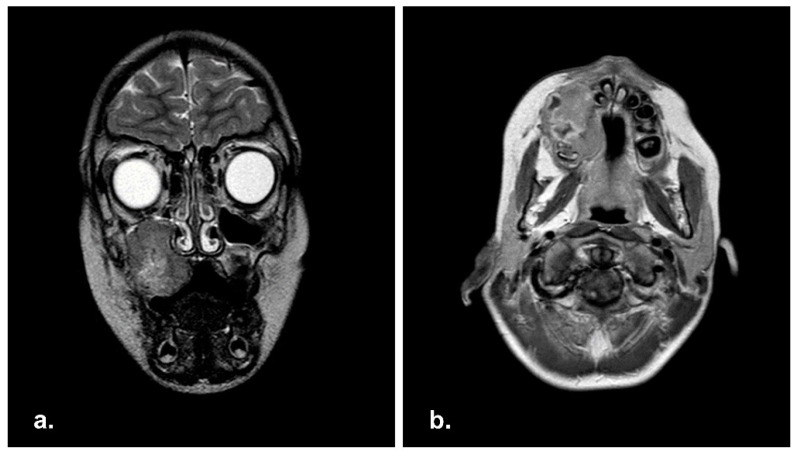
Pre-operative Magnetic Resonance Imaging (MRI) in coronal (**a**) and axial (**b**) view showing a well-demarcated mass in right maxillary sinus with extension in the right hard palate.

**Figure 2 reports-06-00016-f002:**
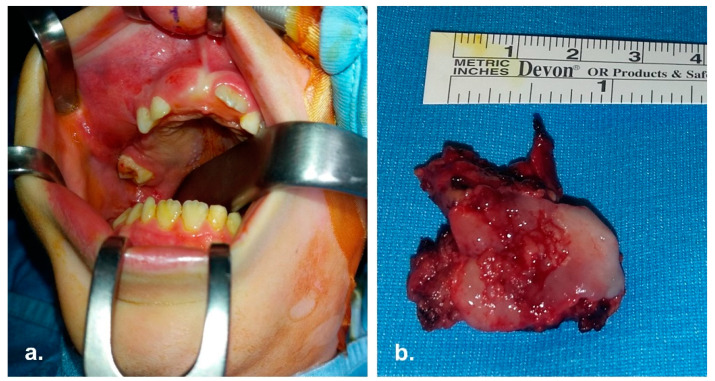
Intra operative pictures. (**a**) Surgical field; the mass involving right maxilla is evident. (**b**) En-bloc enucleation of the mass.

**Figure 3 reports-06-00016-f003:**
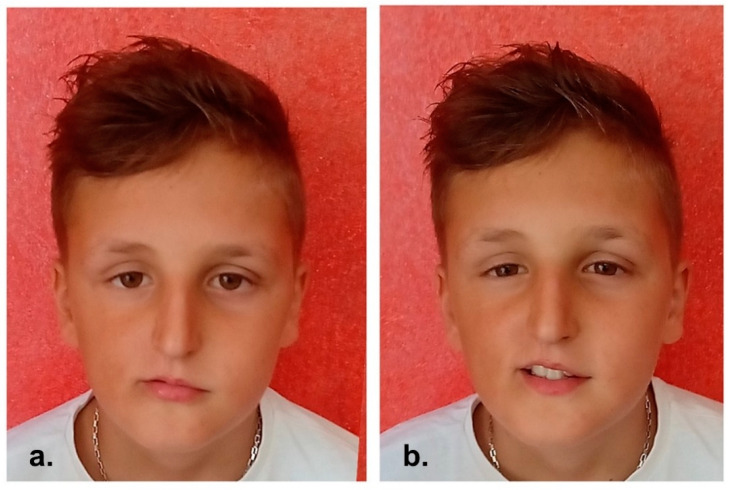
(**a**,**b**) 4-year post-operative clinical pictures.

**Figure 4 reports-06-00016-f004:**
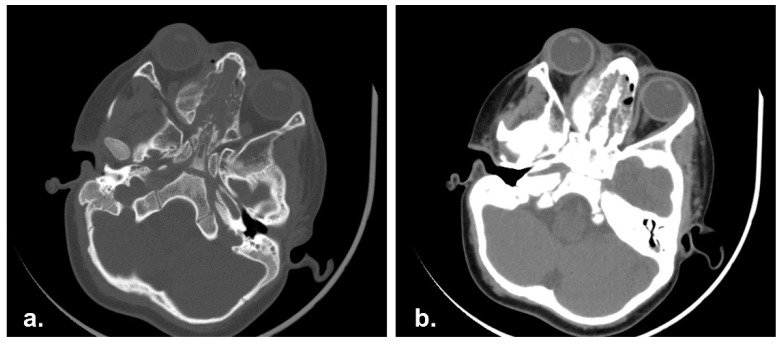
Pre-operative Computed Tomography scan (CT) showing an ethmoidal mass involving the nasal septum and bilateral lamina papyracea. (**a**) Bone and (**b**) soft tissue CT algorithm.

**Figure 5 reports-06-00016-f005:**
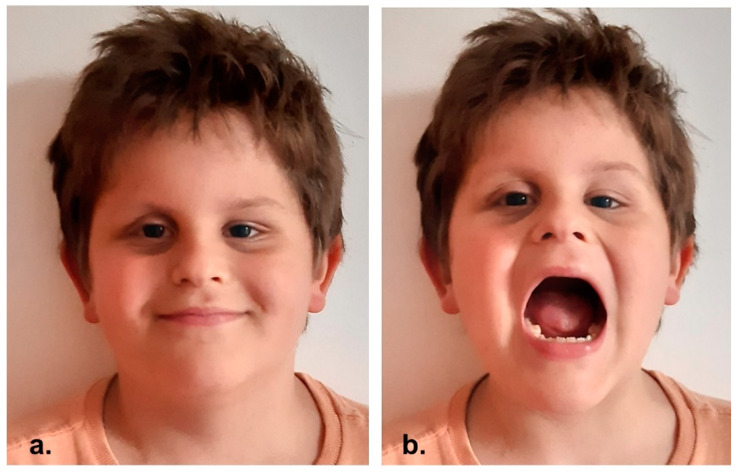
(**a**,**b**) Clinical appearance at 6-years follow-up.

**Figure 6 reports-06-00016-f006:**
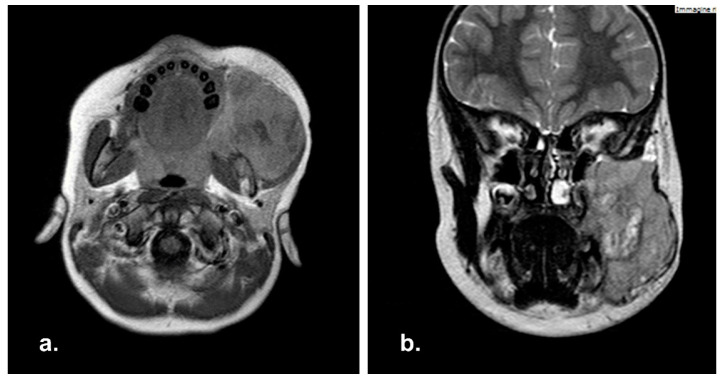
Pre-operative Magnetic Resonance Imaging (MRI) in axial (**a**) and coronal (**b**) view with showing a massive neoplasia in the right masticatory space.

**Figure 7 reports-06-00016-f007:**
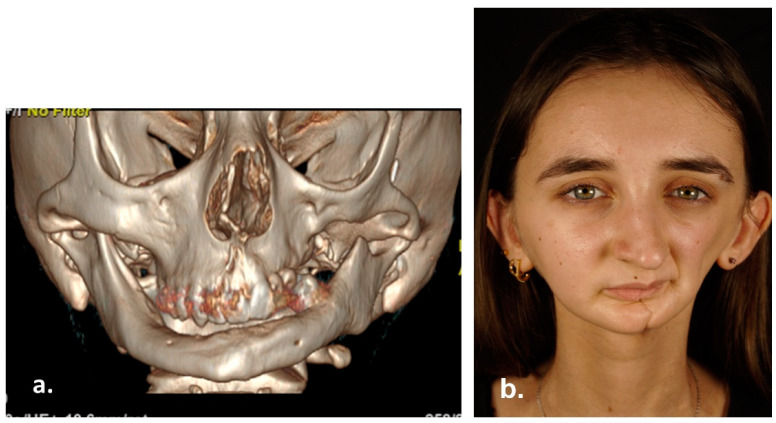
3D Computed Tomography (**a**) and clinical appearance (**b**) 15 years after surgery.

**Figure 8 reports-06-00016-f008:**
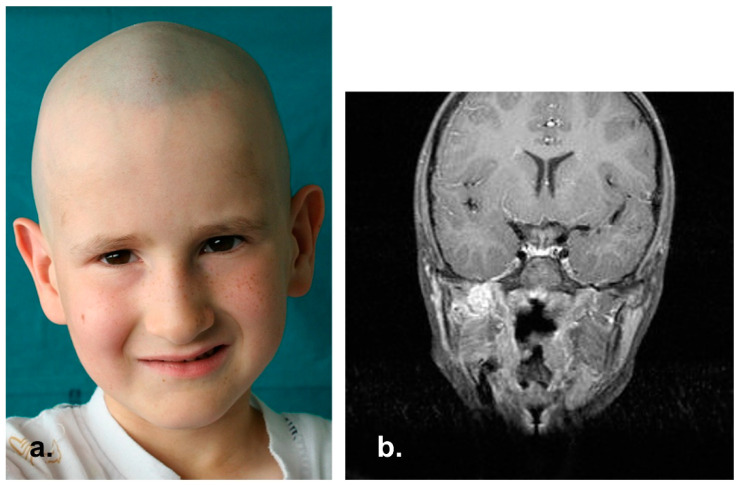
Rhabdomyosarcoma relapse in right masticatory space. (**a**) Clinical picture. (**b**) MRI coronal view showing an enhanced mass in the right pterygopalatine fossa after contrast administration.

**Figure 9 reports-06-00016-f009:**
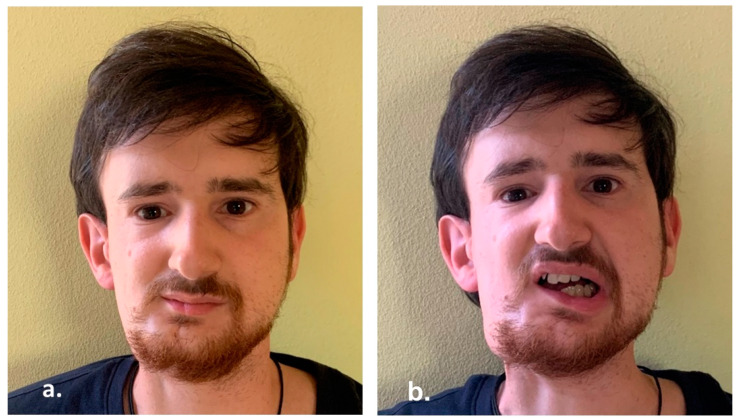
(**a**,**b**) Clinical appearance 15 years after treatments.

**Table 1 reports-06-00016-t001:** Demographic features of study population.

Demographic Features		All Patients (n = 4)
Mean age at diagnosis (year)		4.93 ± 1.58
Mean time since the end of treatment (year)		8.89 ± 5.57
Mean age at clinical evaluation of this study (year)		15.37 ± 6.33
Sex	Man	3
	Female	1

**Table 2 reports-06-00016-t002:** Clinical-pathological characteristics of study population.

Clinical-Pathological Features	All Patients (n = 4)
Histologic Type	
Rhabdomyosarcoma	2
Ewing’s Sarcoma	2
Primary Site	
Maxilla	1
Nose and paranasal sinuses	1
Infratemporal fossa/pterygopalatine fossa	2
Tumor extension at diagnosis	
<5 cm	2
≥5 cm	2

**Table 3 reports-06-00016-t003:** Item score distributions for the Pediatric Quality of Life Inventory (PedsQL) 4.0 Generic Core Scales.

Scale	N° of Item	N° of Patient	Mean	SD
Self-report				
Total Score	23	3	82.92	10.67
Psychosocial health	15	3	82.44	13.02
Physical health	8	3	84.37	8.27
Emotional functioning	5	3	69	22.5
Social functioning	5	3	95	8.66
School functioning	5	3	83.33	12.58

**Table 4 reports-06-00016-t004:** Multi-item score distributions for EORTC QLQ-H&N 43.

Multi-Item Scale	N° of Item	N° of Patient	Mean	SD
Pain in the mouth	4	4	10.41	12.5
Swallowing	4	4	2.08	4.16
Problems with teeth	3	4	33.33	39.5
Dry mouth and sticky saliva	2	4	4.17	8.33
Problems with senses	2	4	0	0
Speech	5	4	25	25.75
Body Image	3	4	16.66	26.45
Social-eating	4	4	8.33	11.8
Sexuality	2	1	0	0
Problems with shoulder	2	4	0	0
Skin Problems	3	4	5.55	11.11
Fear of progression	2	4	12.5	8.33

**Table 5 reports-06-00016-t005:** Single-item score distributions for EORTC QLQ-H&N 43.

Single-Item Scale	N° of Patient	Mean	SD
Problems opening mouth	4	50.00	33.33
Coughing	4	0	0
Social contact	4	16.67	33.33
Swelling in the neck	4	0	0
Weight loss	4	0	0
Problems with wound healing	4	8.33	16.67
Neurological problems	4	0	0

## Data Availability

No new data were created or analyzed in this study. Data sharing is not applicable to this article.

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
