# Peer review of "Long-Term Outcome and Quality of Life in Patients Treated for Head and Neck Sarcoma during Pediatric Age: Considerations from a Series of 4 Cases"

_reports, 2023, doi:10.3390/reports6010016_

Round 1
Reviewer 1 Report
This is a case series of 4 head and neck sarcomas including both, soft tissue and bone sarcomas in children. Unfortunately the authors have broadened the case series descriptive design by employing some analytic methods. I am afraid that there’s no possibility to draw any reliable conclusions from the analysis of their heterogeneous and extremely small study group. Therefore transforming this paper into a descriptive case series focused on quality of life after extensive surgery in head and neck sarcoma patients, without any survival analysis, is recommendable. Achieving a satisfactory health-related quality of life in the analyzed patients may prompt the decision for radical surgery in others.
Besides this major comment I have also some minors
- the names of cytostatics should be in English
- were the negative surgical margins achieved in pts #3 and #4?
- pt #3 – what type of RMS was diagnosed?
- abstract line 25-26: “we described our experience with a series of patients diagnosed with head and neck non-metastatic sarcoma…” – patient # 4 was diagnosed as metastatic.
- lines 39-40: neoplasm in general constitute the second leading cause of death in children and adolescents, not malignant tumors of the head and neck only.
- in the discussion, the authors should also emphasize the impact of the surgical center experience in pediatric oncology on patient outcome.
Author Response
Dear Reviewer,
We thank you for your suggestions, which allowed us to improve our paper. Our corrections and comments are reported below:
Comment:
This is a case series of 4 head and neck sarcomas including both, soft tissue and bone sarcomas in children. Unfortunately, the authors have broadened the case series descriptive design by employing some analytic methods. I am afraid that there’s no possibility to draw any reliable conclusions from the analysis of their heterogeneous and extremely small study group. Therefore, transforming this paper into a descriptive case series focused on quality of life after extensive surgery in head and neck sarcoma patients, without any survival analysis, is recommendable. Achieving a satisfactory health-related quality of life in the analyzed patients may prompt the decision for radical surgery in others.
Answer:
We agree that the quality of life is the main topic of our case series. Anyway, the survival analysis is an adjunctive feature that not misguide the item and, moreover, it results essential in oncological papers. So, we thank the reviewer for the recommendation but we leave the final decision to the Editor.
Comment:
the names of cytostatics should be in English.
Answer:
We have corrected.
Comment:
were the negative surgical margins achieved in pts #3 and #4?
Answer:
We have specified it in the main text.
Comment:
pt #3 – what type of RMS was diagnosed?
Answer:
We have specified it in the main text.
abstract line 25-26:
Comment:
“we described our experience with a series of patients diagnosed with head and neck non-metastatic sarcoma…” – patient # 4 was diagnosed as metastatic.
Answer:
We have clarified and modified the sentence in the main text as follow: “The finding of a pulmonary nodule, considered the local extension of the primitive head and neck tumor, initially led us to a diagnosis of pulmonary metastasis. However, multidisciplinary post-treatment evaluation based on clinical and radiological findings, prompted us to correct the initial diagnosis to non-metastatic rhabdomyosarcoma”.
lines 39-40:
Comment:
neoplasm in general constitute the second leading cause of death in children and adolescents, not malignant tumors of the head and neck only.
Answer:
We have corrected it in the text.
Comment:
In the discussion, the authors should also emphasize the impact of the surgical center experience in pediatric oncology on patient outcome.
Answer:
Thank you for your suggestion. We have enriched the discussion, with related references, as follows: “Complete surgical resection is beneficial for HNRMS and is an important aspect of the multimodal management protocol endorsed by the COG. However, depending on tumor site, complete tumor resection can be challenging. It is necesaary that the patients are reffered to a center with high experience in pediatric oncological surgery”.
Reviewer 2 Report
The authors describe 4 of their 6 children with head and neck sarcoma treated at their institution over the period 1995 -2020.
General remarks:
Even if these are only very few cases (which is due to the low prevalence of the specific disease) the reported observations are worth reading.
However, perhaps the number of patients studied should already be mentioned in the title.
Introduction:
The authors write in the Introduction that pediatric sarcomas are divided into soft tissue sarcomas and bone sarcomas. Of course, this is also true for sarcomas in adults. The sentence can therefore be formulated more generally
Materials and methods:
Two QoL questionnaires were completed by the 4 patients. PedsQL 4.0 and EORTC QLQ-H&N 43. For the interpretation of the PedsQL 4.0 results, the period of acquisition should be provided to understand the conclusions that the COVID pandemic (lockdown) may have affected the responses.
Results:
The 4 patients are impressively illustrated.
In the first case, a second radiation therapy is described, but no information is given about a first radiation therapy.
In case 3, the correct English names of the drugs (generics) are missing.
Table 1. Demographic features of study population: Are numbers given as median or mean for “age at diagnosis” and “age at clinical evaluation” ?
References:
Not all the cited references are relevant to this research (e.g. Ref 26 about the COVID situation at the institution which is from 2020 whereas the current study probably dates from 2021, according to project identification code 1/2021 )
Reference 2: Comprehensive Cancer Information - National Cancer Institute. Do you have a specific URL address, you would like to mention ?
Reference 17: Protocol EpSSG RMS 2005 Final 1.1 16.09.05_1_.Doc.
Any open access to the protocol ? If not, please Insert the protocol version into the text
Author Response
Dear Reviewer,
We thank you for your suggestions, which allowed us to improve our paper. Our corrections and comments are reported below:
General remarks:
Comment:
Even if these are only very few cases (which is due to the low prevalence of the specific disease) the reported observations are worth reading.
However, perhaps the number of patients studied should already be mentioned in the title.
Answer:
We have modified the title as request.
Introduction:
Comment:
The authors write in the Introduction that pediatric sarcomas are divided into soft tissue sarcomas and bone sarcomas. Of course, this is also true for sarcomas in adults. The sentence can therefore be formulated more generally
Answer:
We have modified the sentence.
Materials and methods:
Comment:
Two QoL questionnaires were completed by the 4 patients. PedsQL 4.0 and EORTC QLQ-H&N 43. For the interpretation of the PedsQL 4.0 results, the period of acquisition should be provided to understand the conclusions that the COVID pandemic (lockdown) may have affected the responses.
Answer:
We fully agree with the suggestion, thank you. We have specified the period of acquisition of the QoL questionnaires in the main text.
Results:
Comment:
The 4 patients are impressively illustrated.
In the first case, a second radiation therapy is described, but no information is given about a first radiation therapy.
Answer:
The sentence was misleading. We have clarified and modified it.
Comment:
In case 3, the correct English names of the drugs (generics) are missing.
Answer:
Corrected.
Comment:
Table 1. Demographic features of study population: Are numbers given as median or mean for “age at diagnosis” and “age at clinical evaluation” ?
Answer:
Thank you. We have corrected the table.
References:
Comments:
Not all the cited references are relevant to this research (e.g. Ref 26 about the COVID situation at the institution which is from 2020 whereas the current study probably dates from 2021, according to project identification code 1/2021 )
Answer:
The present study was certainly conducted in 2021. However, the QoL questionnaires were administered to patients included in this study during the period from October 2020 to April 2021, when the Covid-19 pandemic emergency was still ongoing. Social activities in that period were still subject to the limitations dictated by the legislation in force, in Italy. We believe that this contingent variable may have contributed to our results in terms of quality of life, and should therefore be taken into consideration. For this reason, we do not deem the citation indicated as inappropriate for the present research.
Reference 2: Comprehensive Cancer Information - National Cancer Institute. Do you have a specific URL address, you would like to mention ?
Answer:
We have added the URL address.
Reference 17: Protocol EpSSG RMS 2005 Final 1.1 16.09.05_1_.Doc.
Any open access to the protocol ? If not, please Insert the protocol version into the text
Answer:
Done, thank you.
Round 2
Reviewer 1 Report
I thank the authors for consideration of my previous comments. Nevertheless, I maintain my previous comment that a survival analysis of such a small and heterogeneous study group in terms of diagnosis and treatment modalities is unjustified.
Author Response
Comment:
I thank the authors for consideration of my previous comments. Nevertheless, I maintain my previous comment that a survival analysis of such a small and heterogeneous study group in terms of diagnosis and treatment modalities is unjustified.
Answer:
Thank you for the suggestion. We have deleted the paper section “survival outcome”, while the overall survival rate was mentioned as generic result. We included the survival curve as supplementary material in the corresponding section of the paper. Additionally, in the abstract and material and methods section we have modified the overall survival from a main purpose into a general result of our study.
We hope that this revised version of the manuscript will match your recommendations.

Reviewer 2 Report
please correct: line 215
anatomical-pathological examination or anatomic-pathological examination
Round 3
Reviewer 1 Report
no further comments